# Empirical modeling of the percent depth dose for megavoltage photon beams

**Xiao-Jun Li**[1][*], **Yan-Cheng Ye**[1], **Yan-Shan Zhang**[1‡], **Jia-Ming Wu**[1,2,3‡*]

**1** Heavy Ion Center of Wuwei Cancer Hospital; Gansu Wuwei Academy of Medical Sciences; Gansu Wuwei Tumor Hospital, Wuwei City, Gansu province, China, **2** Department of Medical Physics, Chengde Medical University, Chengde City, Hebei Province, China, **3** Department of Radiation Oncology, Yee Zen General Hospital, Tao Yuan City, Taiwan

◉ These authors contributed equally to this work.
‡ YSZ and JMW authors also contributed equally to this work.
* anglweli@qq.com (XJL); jiaming.wu@chmsc.com (JMW)

**Data Availability Statement:** All relevant data are within the paper and supporting information.

## Abstract

### Introduction

This study presents an empirical method to model the high-energy photon beam percent depth dose (PDD) curve by using the home-generated buildup function and tail function (buildup-tail function) in radiation therapy. The modeling parameters n and μ of buildup-tail function can be used to characterize the Collimator Scatter Factor ($S_c$) either in a square field or in the different individual upper jaw and lower jaw setting separately for individual monitor unit check.

### Methods and materials

The PDD curves for four high-energy photon beams were modeled by the buildup and tail function in this study. The buildup function was a quadratic function in the form of $\frac{d}{\sqrt{d^2+n}}$ with the main parameter of d (depth in water) and n, while the tail function was in the form of $e^{-\mu d}$ and was composed by an exponential function with the main parameter of d and μ. The PDD was the product of buildup and tail function, PDD $= \frac{d}{\sqrt{d^2+n}} \cdot e^{-\mu d}$. The PDD of four-photon energies was characterized by the buildup-tail function by adjusting the parameters n and μ. The $S_c$ of 6 MV and 10 MV can then be expressed simply by the modeling parameters n and μ.

### Results

The main parameters n increases in buildup-tail function when photon energy increased. The physical meaning of the parameter n expresses the beam hardening of photon energy in PDD. The fitting results of parameters n in the buildup function are 0.17, 0.208, 0.495, 1.2 of four-photon energies, 4 MV, 6 MV, 10 MV, 18 MV, respectively. The parameter μ can be treated as attenuation coefficient in tail function and decreases when photon energy increased. The fitting results of parameters μ in the tail function are 0.065, 0.0515, 0.0458, 0.0422 of four-photon energies, 4 MV, 6 MV, 10 MV, 18 MV, respectively. The values of n

**Funding:** This work was supported by the Funding: Key R&D plan of Science and Technology Program of Gansu Province, China. (19YF3FH001).

**Competing interests:** The authors have declared that no competing interests exist.

and μ obtained from the fitted buildup-tail function were applied into an analytical formula of $S_c = n_E(S)^{0.63\mu_E}$ to get the collimator to scatter factor $S_c$ for 6 and 10 MV photon beam, while $n_E$, $\mu_E$, S denotes n, μ at photon energy E of field size S, respectively. The calculated $S_c$ were compared with the measured data and showed agreement at different field sizes to within ±1.5%.

## Conclusions

We proposed a model incorporating a two-parameter formula which can improve the fitting accuracy to be better than 1.5% maximum error for describing the PDD in different photon energies used in clinical setting. This model can be used to parameterize the $S_c$ factors for some clinical requirements. The modeling parameters n and μ can be used to predict the $S_c$ in either square field or individual jaws opening asymmetrically for treatment monitor unit double-check in dose calculation. The technique developed in this study can also be used for systematic or random errors in the QA program, thus improves the clinical dose computation accuracy for patient treatment.

## I. Introduction

The measurement of x-ray dose at the central axis in radiation oncology is usually tabulated and used for the clinical dose calculation. Percentage depth dose (PDD) and tissue-phantom ratio (TPR) are dominated by the scattering effect of depth and field size [1]. They consist mainly in two parts, primary fluence (adjusted for inverse square, beam hardening effects) and another part, which represents the effects of attenuation. PDD and the other quantities such as TPR and field size factor ($S_{cp}$) are typically measured for simple square-shaped fields for each therapy machine and modality [2, 3].

The collection of clinical data for the implantation of treatment planning system for dose accuracy calculation is time consuming and repetitive. The achievement of this study provides a lot of help in double check of the measurement results to reduce the time of measurements, and the confidence to use interpolation. It would be helpful to know the smallest number of measurements needed to characterize the high-energy x-ray beam [4]. Several protocol provide a general framework and describes a large number of tests and procedures that should be considered by the users of RTPSs. However, the workload for the implementation of the recommendations from those documents are enormous and requires far more personnel and instrumentation resources than is available in most facilities, particularly within smaller hospitals. These hospitals are not always able to perform complete characterization, algorithm validation and software testing of dose calculation algorithms used in RTPS [5]. These may include the $S_c$ factors, also known as head scatter factors, which account for the variation in beam output with field size from changes in direct and indirect radiation from the head of the linear accelerator. The term $S_{cp}$ contains both the collimator and phantom scatter that is defined as the ratio of dose for the field of interest to that of a reference field for the same delivered monitor units measured under full scatter conditions in a large water tank at the reference depth [6].

In this study, we proposed a simple mathematic equation to model the PDD by using the buildup-tail function. The quantity of interest is the parameters extracted from the well modeled PDD buildup-tail function for characterizing the $S_c$ either in a square field or in the

different individual upper jaw and lower jaw setting separately for specific patient treatment monitor unit calculation.

## II. Materials and methods

### II. A Experiment design and steps

The experiment was conducted in the following steps:

1. The PDD for four high-energy photon energies of 4 MV, 6 MV, 10 MV, 18 MV were modeled in this study. The measurements data were 6 MV, 10 MV (Elekta infinity, Stockholm, Sweden, and **Varian trueBeam,** Palo Alto, Ca), while 4 MV and 18 MV were published data [7, 8].

2. Measurement of photon beam PDD was conducted by two linear accelerators with two-photon energies of 6 MV and 10 MV at SSD = 100cm with different field sizes. The quantity percentage depth dose is defined as the quotient, expressed as a percentage, of the absorbed dose at any depth d to the absorbed dose at a reference depth (usually at the depth of dose maximum) along the central axis of the beam.

3. The $S_c$ for radiation beams of two-photon energies of 6 MV and 10 MV were measured.

4. High energy photon PDD curves were modeled by the buildup and tail function generated in this study.

5. The PDD of four-photon energies were modeled by the buildup-tail function by adjusting the parameters n and μ to get the best fitting.

6. The $S_c$ of 6 MV and 10 MV can then be expressed simply by the modeling parameters n and μ.

   The details of each step are described in the following sections.

### II. B Percent depth dose measurement

PDDs were acquired with a PTW MP3-T water phantom (PTW ionization Freiburg Gmbh) at **W**uWei **H**eavy **I**on **C**enter, Wuwei **C**ancer **H**ospital, Gansu, China (**WHICH**). PDDs were measured with a PTW Semiflex parallel-plate ionization chamber (PTW ionization Freiburg Gmbh, type 31010, volume 0.125 $cm^3$). For the acquisition of PDDs the chamber position was automatically corrected to the effective point of measurement. In both photon energies water phantom measurements were performed at 100 cm source-to-surface distance (SSD) for square field sizes of 5 x 5, 10 x 10, 15 x 15, 20 x 20, and 40 x 40 $cm^2$ with a step size of 0.1 cm. PDDs were normalized to 100% at $d_{max}$ depth. Since the parallel-plate chamber has a small plate separation and it is explicit that the point of measurement is the front surface of the cavity. The depth curve measured by parallel-plate chamber was then compared with the PDD curve measured by films.

### II. C The comparison of depth dose curve converted via parallel-plate chamber ionization curve with GAF EBT 3film

We used Gafchromic EBT3 films (Ashland Specialty Ingredients GP, NJ USA; Lot # 04022001) for the depth dose curves measurement in for comparing the PDD measured by plane-parallel ion chamber. The film processing and dose profile measurements followed the international protocols [9]. A pre-exposure technique was used for the calibration curve derivation [10]. This was performed by giving each film an initial dose of 2 Gy with a photon energy of 6 MV

to homogenize the film polymer density in Gafchromic film experiment. The Hurter-Driffield calibration curve (H-D curve) [11] and the PDD films data scanned with a red filter [12] were analyzed by the VeriSoft imaging procession software (PTW-Freiburg, Germany, VeriSoft version 3.1) for further comparing with the depth dose curves measured by the parallel-plate chamber.

Absolute output and machine quality assurance were performed before conducting the measurements of percent ionization depth by the parallel-plate chamber as well as PDD curve by Gafchromic films.

## II.D Measurement of Sc

The measurement of $S_c$ for Elekta infinity and Varian trueBeam with the high-energy photon energy of 6 and 10 MV were conducted using a PTW Semiflex chamber, type31010 (PTW, Freiburg), coupled to either a PTW UNIDOS or a Scanditronix-Wellhofer Dose1 (IBADosimetry, Schwarzenbruck) electrometer. The chamber and phantom fulfill the suitability criteria for this type of measurement according to the recommendation by the AAPM TG-74 report. The chamber was placed in an aluminum($\rho$ = 2.7 gcm$^3$) mini phantom with 3.9 cm of the material above the chamber, a measurement depth beyond $d_{max}$ equivalent to a depth of 10 cm water for sufficiently avoiding contaminant electrons. The phantom and chamber axis were vertically aligned to the beam central axis, and the chamber reference point was set at 100 cm source-to-chamber distance. All measurements were normalized to the reference field reading of 10 x10 cm$^2$. $S_c$ values were measured for a selection of square fields from 3 x 3 to 40 x 40 cm$^2$ with the chamber in mini-phantom.

## II. E percent depth dose numerical equation

There are two home-generated numerical equations for describing the PDD curves of high energy photon beam, buildup function, and tail function. The buildup function was a quadratic function in the form of $\frac{d}{\sqrt{d^2+n}}$ with two main parameters of d (depth in water) and n, while the tail function was in the form of $e^{-\mu d}$ and was composed by an exponential function with main parameters of d and $\mu$. The modeled PDD was the product of buildup and tail function to be, PDD$_{b-t}$ = $\frac{d}{\sqrt{d^2+n}} \cdot e^{-\mu d}$.

PDD$_{b-t}$ is described separately as buildup function and tail function in the following,

$$\text{buildup function}: \quad \frac{d}{\sqrt{d^2+n}},$$

where d is the depth in water along the central axis in unit cm, n is a beam hardening factor with unitless, a scalar.

$$\text{tail function}: \quad e^{-\mu d},$$

where d is the depth in water along the central axis in unit cm, $\mu$ is a linear attenuation coefficient factor in unit cm$^{-1}$ for adjusting the slope of the tail. The tail function in the form of $e^{-\mu d}$ was composed of an exponential function with main parameters of d and $\mu$.

The empirical function of percentage depth dose is the combination of these two functions, denoted as PDD$_{b-t}$ (the abbreviation of PDD$_{buildup-tail}$)

$$\text{PDD}_{b-t} = \frac{d}{\sqrt{d^2+n}} \cdot e^{-\mu d} \qquad \text{Eq1}$$

The PDD of four photon energies were modeled by the buildup-tail function by adjusting the parameters n and μ to get the best fitting.

All PDD of four high-energy photon beams with different field sizes at SSD = 100cm were adjusted by the main parameters of n and μ to get the best fitting.

## III. Results

### III. A. The best fitting of percent depth dose was conducted by empirical function in four-photon energies

The dose variations of PDD of high energy photon beams measured by films and by ion chamber were less than 0.5%.

The PDD with different energies adopted in this study was already measured by the water phantom during commissioning and was spot checked in this experiment. By adjusting the main parameters of n and μ, the best fitting for four-photon PDD curves in every energy at the field sizes of 10 cm x 10 cm were listed in Fig 1A–1D. The comparison of four high-energy photon measured and modeled PDD with two published and two facility data is listed in Table 1.

Fig 1A–1D represents the fitting results of PDD curves of photon energy from 4 to 18 MV, representatively. Table 2 lists the best fitting parameters n and μ for four-photon energies.

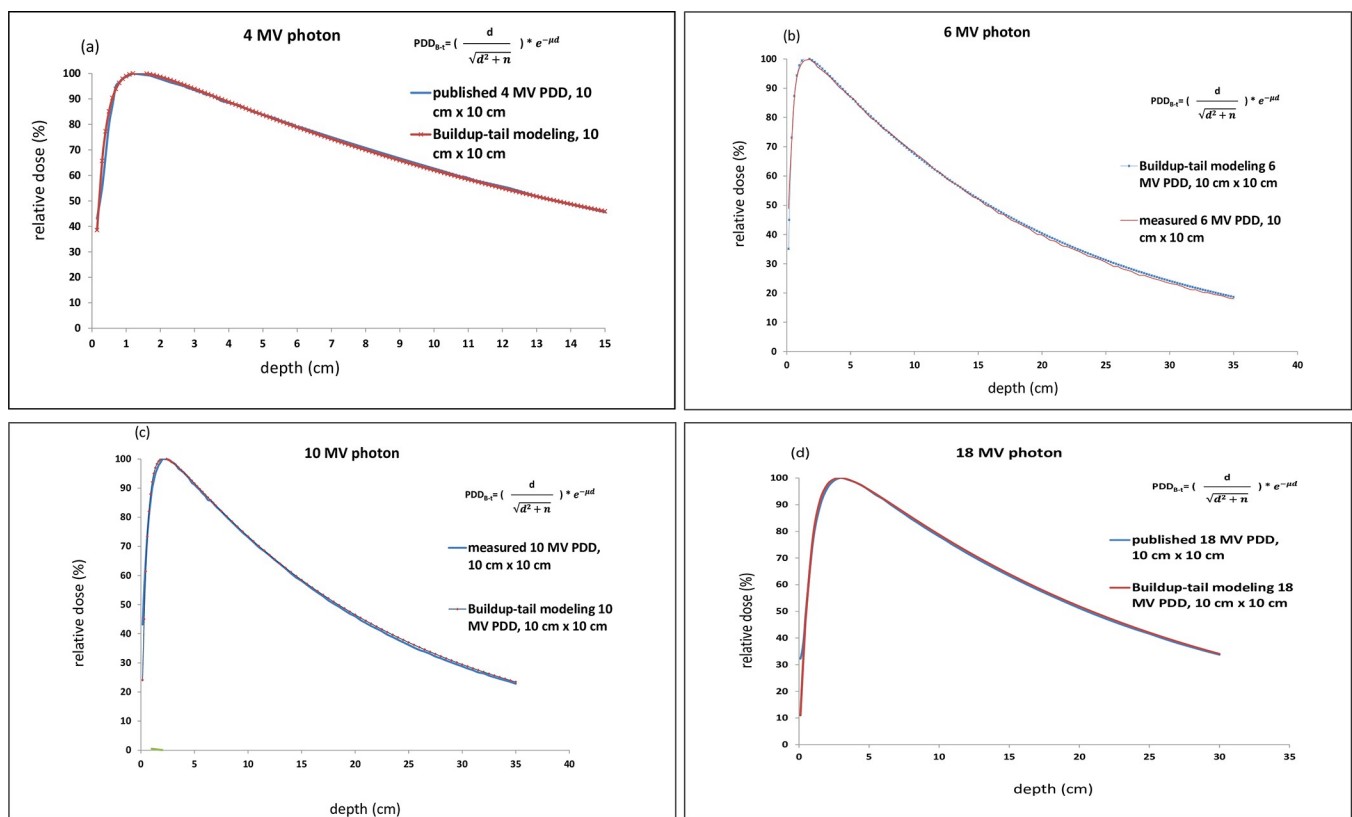

**Fig 1.** The best fitting of photon PDD curve at energy 4, 6, 10 and 18 MV from a–d), representatively. The measurements data were 6 MV (b) and 10 MV (c) while 4 MV (a) and 18 MV (d) were published data.

**Table 1. The comparison of four high-energy photon measured and modeled PDD with two facility and two published data.**

| depth (cm) | 4 MV photon (published data) | | | 6 MV photon facility data) | | | 10 MV photon (facility data) | | | 18 MV photon (published data) | | |
|---|---|---|---|---|---|---|---|---|---|---|---|---|
| | measured PDD | modeled PDD | error (%) | measured PDD | modeled PDD | error (%) | measured PDD | modeled PDD | error (%) | measured PDD | modeled PDD | error (%) |
| 0.1 | 38.89 | 29.46 | -24.24 | 40.12 | 23.28 | -41.97 | 38.98 | 17.00 | -56.39 | 32.20 | 10.94 | -66.03 |
| 1 | 98.67 | 98.75 | 0.09 | 98.09 | 97.12 | -0.99 | 93.91 | 90.70 | -3.42 | 76.55 | 78.09 | 2.01 |
| 1.5 | 98.80 | 98.92 | 0.12 | 100.08 | 100.08 | 0.00 | 97.39 | 95.82 | -1.62 | 89.55 | 91.59 | 2.27 |
| 2 | 98.44 | 98.59 | 0.15 | 99.62 | 99.56 | -0.07 | 100.02 | 99.80 | -0.22 | 96.07 | 97.39 | 1.38 |
| 2.5 | 96.17 | 96.32 | 0.16 | 98.08 | 97.82 | -0.27 | 99.22 | 99.24 | 0.02 | 99.01 | 99.59 | 0.58 |
| 3 | 93.78 | 93.93 | 0.16 | 96.10 | 95.66 | -0.45 | 97.96 | 98.23 | 0.27 | 100.00 | 100.00 | 0.00 |
| 3.5 | 91.21 | 91.36 | 0.16 | 93.89 | 93.33 | -0.59 | 96.01 | 96.52 | 0.53 | 99.33 | 99.48 | 0.15 |
| 4 | 88.65 | 88.80 | 0.17 | 91.59 | 90.95 | -0.70 | 94.04 | 94.63 | 0.63 | 98.42 | 98.43 | 0.01 |
| 4.5 | 86.10 | 86.25 | 0.17 | 89.25 | 88.56 | -0.78 | 92.04 | 92.62 | 0.63 | 97.21 | 97.09 | -0.12 |
| 5 | 83.61 | 83.76 | 0.18 | 86.92 | 86.19 | -0.84 | 90.01 | 90.57 | 0.61 | 95.40 | 95.57 | 0.18 |
| 5.5 | 81.14 | 81.29 | 0.18 | 84.62 | 83.86 | -0.89 | 87.98 | 88.50 | 0.59 | 93.58 | 93.95 | 0.39 |
| 6 | 78.78 | 78.93 | 0.19 | 82.35 | 81.58 | -0.93 | 85.97 | 86.44 | 0.55 | 92.09 | 92.27 | 0.20 |
| 6.5 | 76.44 | 76.59 | 0.20 | 80.12 | 79.35 | -0.97 | 83.87 | 84.40 | 0.64 | 90.15 | 90.56 | 0.46 |
| 7 | 74.19 | 74.34 | 0.20 | 77.94 | 77.17 | -0.99 | 81.38 | 82.40 | 1.25 | 88.34 | 88.84 | 0.57 |
| 7.5 | 71.99 | 72.14 | 0.21 | 75.82 | 75.04 | -1.02 | 79.14 | 80.43 | 1.63 | 86.57 | 87.12 | 0.64 |
| 8 | 69.85 | 70.00 | 0.21 | 73.74 | 72.97 | -1.04 | 77.27 | 78.50 | 1.58 | 84.72 | 85.41 | 0.82 |
| 8.5 | 67.78 | 67.93 | 0.22 | 71.71 | 70.96 | -1.05 | 75.44 | 76.60 | 1.54 | 83.05 | 83.71 | 0.80 |
| 9 | 65.75 | 65.90 | 0.23 | 69.74 | 68.99 | -1.07 | 73.64 | 74.74 | 1.50 | 81.31 | 82.04 | 0.89 |
| 9.5 | 63.81 | 63.96 | 0.24 | 67.81 | 67.08 | -1.08 | 71.89 | 72.93 | 1.45 | 79.68 | 80.39 | 0.89 |
| 10 | 61.89 | 62.04 | 0.24 | 65.94 | 65.22 | -1.09 | 70.16 | 71.14 | 1.40 | 78.07 | 78.76 | 0.88 |
| 10.5 | 60.06 | 60.21 | 0.25 | 64.12 | 63.41 | -1.10 | 68.50 | 69.43 | 1.35 | 76.52 | 77.16 | 0.83 |
| 11 | 58.26 | 58.41 | 0.26 | 62.34 | 61.65 | -1.10 | 66.85 | 67.72 | 1.31 | 74.83 | 75.58 | 1.01 |
| 11.5 | 56.54 | 56.69 | 0.27 | 60.61 | 59.94 | -1.11 | 65.26 | 66.08 | 1.26 | 73.31 | 74.03 | 0.99 |
| 12 | 54.84 | 54.99 | 0.27 | 58.93 | 58.27 | -1.12 | 63.69 | 64.46 | 1.21 | 71.74 | 72.52 | 1.07 |
| 12.5 | 53.20 | 53.35 | 0.28 | 57.30 | 56.66 | -1.12 | 62.17 | 62.89 | 1.16 | 70.28 | 71.04 | 1.08 |
| 13 | 51.58 | 51.73 | 0.29 | 55.71 | 55.08 | -1.13 | 60.68 | 61.35 | 1.11 | 68.81 | 69.56 | 1.09 |
| 13.5 | 49.96 | 50.11 | 0.30 | 54.16 | 53.55 | -1.13 | 59.21 | 59.84 | 1.06 | 67.40 | 68.14 | 1.10 |
| 14 | 48.35 | 48.50 | 0.31 | 52.66 | 52.06 | -1.14 | 57.80 | 58.38 | 1.01 | 65.99 | 66.72 | 1.11 |
| 14.5 | 46.73 | 46.88 | 0.32 | 51.20 | 50.61 | -1.14 | 56.39 | 56.93 | 0.96 | 64.65 | 65.35 | 1.09 |
| 15 | 45.11 | 45.26 | 0.33 | 49.77 | 49.20 | -1.14 | 55.05 | 55.55 | 0.91 | 63.30 | 63.99 | 1.08 |
| 15.5 | 43.50 | 43.65 | 0.34 | 48.39 | 47.84 | -1.15 | 53.72 | 54.18 | 0.86 | 61.99 | 62.68 | 1.10 |
| 16 | 41.88 | 42.03 | 0.36 | 47.04 | 46.50 | -1.15 | 52.44 | 52.86 | 0.81 | 60.68 | 61.36 | 1.13 |
| 16.5 | 40.26 | 40.41 | 0.37 | 45.74 | 45.21 | -1.15 | 51.17 | 51.56 | 0.76 | 59.43 | 60.10 | 1.14 |
| 17 | 38.65 | 38.80 | 0.39 | 44.46 | 43.95 | -1.15 | 49.94 | 50.30 | 0.71 | 58.18 | 58.84 | 1.15 |
| 17.5 | 37.03 | 37.18 | 0.41 | 43.23 | 42.73 | -1.16 | 48.74 | 49.07 | 0.66 | 56.98 | 57.63 | 1.15 |
| 18 | 35.42 | 35.57 | 0.42 | 42.03 | 41.54 | -1.16 | 47.56 | 47.86 | 0.61 | 55.78 | 56.42 | 1.15 |
| 18.5 | 33.80 | 33.95 | 0.44 | 40.86 | 40.38 | -1.16 | 46.43 | 46.69 | 0.56 | 54.61 | 55.26 | 1.19 |
| 19 | 32.18 | 32.33 | 0.47 | 39.72 | 39.26 | -1.16 | 45.30 | 45.53 | 0.51 | 53.45 | 54.10 | 1.23 |
| 19.5 | 30.57 | 30.72 | 0.49 | 38.62 | 38.17 | -1.16 | 44.22 | 44.42 | 0.46 | 52.31 | 52.99 | 1.30 |
| 20 | 28.95 | 29.10 | 0.52 | 37.54 | 37.10 | -1.16 | 43.15 | 43.32 | 0.41 | 51.17 | 51.88 | 1.37 |
| 20.5 | 27.34 | 27.49 | 0.55 | 36.50 | 36.07 | -1.16 | 42.11 | 42.27 | 0.36 | 50.09 | 50.81 | 1.42 |
| 21 | 25.72 | 25.87 | 0.58 | 35.48 | 35.07 | -1.17 | 41.10 | 41.23 | 0.31 | 49.02 | 49.74 | 1.47 |
| 21.5 | 24.10 | 24.25 | 0.62 | 34.49 | 34.09 | -1.17 | 40.11 | 40.21 | 0.26 | 47.99 | 48.71 | 1.50 |
| 22 | 22.49 | 22.64 | 0.67 | 33.53 | 33.14 | -1.17 | 39.14 | 39.23 | 0.21 | 46.97 | 47.69 | 1.53 |
| 22.5 | 20.87 | 21.02 | 0.72 | 32.60 | 32.22 | -1.17 | 38.20 | 38.26 | 0.16 | 46.01 | 46.71 | 1.52 |
| 23 | 19.25 | 19.40 | 0.78 | 31.69 | 31.32 | -1.17 | 37.28 | 37.32 | 0.11 | 45.05 | 45.72 | 1.50 |

**Table 2. This table shows the best fitting parameters n and μ for four-photon energies.**

| parameters | 4 MV | 6 MV | 10MV | 18MV |
|---|---|---|---|---|
| n | 0.17 | 0.208 | 0.495 | 1.2 |
| μ | 0.0605 | 0.0515 | 0.0458 | 0.0422 |

## III. B. The $S_c$ expressed by the parameters n and μ generated in the empirical buildup-tail function of photon energy 6 and 10 MV

The $S_c$ of photon energy 6 and 10 MV can be expressed in a certain acceptable deviation within 0.8% by the parameters n and μ as a function of field size in empirical buildup-tail function by the equation:

$$S_{c,E} = n_E \cdot (FS)^{0.63\mu_E} \qquad \text{Eq2}$$

$n_E$ and $\mu_E$ denote the parameters n and μ in empirical buildup-tail function at photon energy E, while FS stand for field size of interest.

Fig 2A and 2B show the best fitting of the $S_c$ by the $S_{c,E}$ equation.

The measured $S_c$ of two-photon energies can be characterized by the parameters n and μ generated in the buildup-tail function and the comparison between characterized and measured was listed in Table 3.

Table 3 shows the measured $S_c$ in a range of 0.9709 to 1.046 for Varian 6 MV flattened photon beam and 0.9739 to 1.0408 for 10 MV flattened photon beam at square field sizes from 4 cm x 4 cm to 40 cm x 40 cm. The deviation of $S_c$ was characterized by Eq 2 and the measurements were between 0.8% to -0.2% for 6 MV, while for 10 MV were within 0.1%.

## IV. Discussion

To let readers better understand the origin of the home-generated Buildup-tail function. The author describes the derivation of this function in the following.

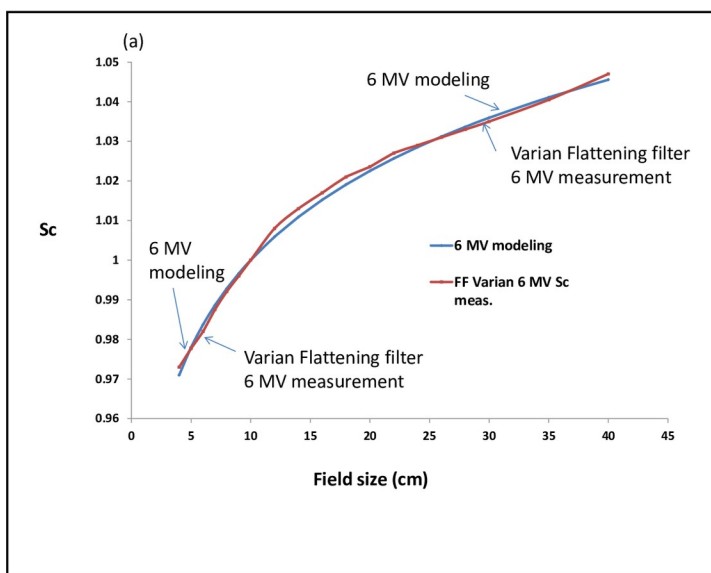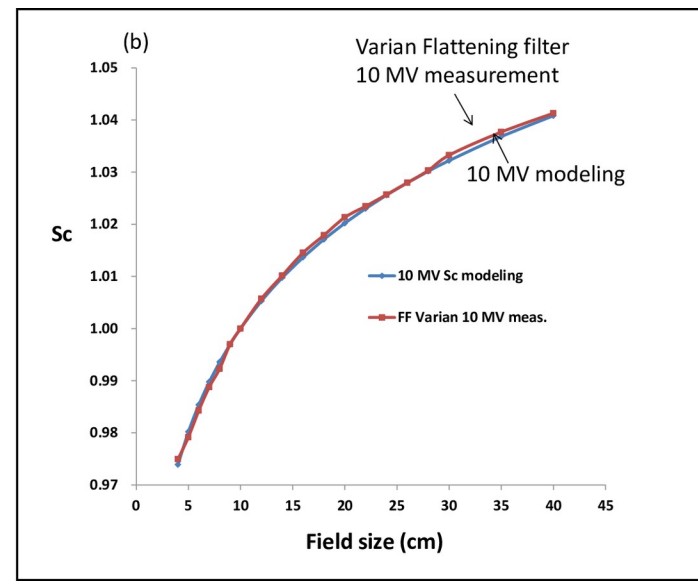

**Fig 2.** The $S_c$ of photon energy 6 MV (a) and 10 MV (b) can be expressed in a certain acceptable deviation within 0.8% by using the parameters n and μ modeled in empirical Buildup-tail function by the equation of $S_{c,E} = n_E \cdot (FS)^{0.63\mu_E}$, $n_E$ and $\mu_E$ denote the parameters n and μ in empirical Buildup-tail function at photon energy E, FS denotes the field size.

**Table 3. This table shows the deviation of calculated and measured $S_c$ at different square field sizes of Varian at photon energy 6 and 10 MV.**

| Field Size (cm x cm) | 6 MV $S_c$ modeling | FF Varian 6 MV measurement | FF 6 MV modeling/meas. error (%) | 10 MV $S_c$ modeling | FF Varian 10 MV measurement | FF 10 MV modeling/meas. error (%) |
|---|---|---|---|---|---|---|
| 4 | 0.9709±0.005 | 0.9690±0.004 | 0.8005±0.002 | 0.9739±0.003 | 0.9670±0.004 | 0.0072±0.003 |
| 5 | 0.9778±0.004 | 0.9727±0.005 | 0.5204±0.003 | 0.9802±0.003 | 0.9742±0.003 | 0.0062±0.003 |
| 6 | 0.9836±0.005 | 0.9810±0.003 | 0.2612±0.002 | 0.9854±0.004 | 0.9808±0.004 | 0.0047±0.002 |
| 7 | 0.9885±0.003 | 0.9875±0.003 | 0.1007±0.002 | 0.9898±0.004 | 0.9869±0.004 | 0.0029±0.002 |
| 8 | 0.9928±0.002 | 0.9921±0.003 | 0.0692±0.003 | 0.9936±0.003 | 0.9923±0.005 | 0.0013±0.003 |
| 9 | 0.9966±0.004 | 0.9960±0.004 | 0.0590±0.003 | 0.9970±0.004 | 0.9970±0.004 | 0.0000±0.004 |
| 10 | 1.0000±0.005 | 1.0000±0.005 | 0.0000±0.004 | 1.0000±0.003 | 1.0000±0.004 | 0.0000±0.003 |
| 12 | 1.0059±0.006 | 1.0080±0.003 | -0.2051±0.002 | 1.0053±0.004 | 1.0071±0.003 | -0.0018±0.003 |
| 14 | 1.0110±0.004 | 1.0130±0.005 | -0.1997±0.003 | 1.0098±0.003 | 1.0120±0.003 | -0.0022±0.004 |
| 16 | 1.0154±0.005 | 1.0170±0.005 | -0.1607±0.002 | 1.0137±0.004 | 1.0160±0.003 | -0.0023±0.003 |
| 18 | 1.0193±0.005 | 1.0210±0.004 | -0.1710±0.003 | 1.0171±0.004 | 1.0201±0.003 | -0.0029±0.004 |
| 20 | 1.0227±0.004 | 1.0236±0.003 | -0.0836±0.002 | 1.0202±0.003 | 1.0234±0.004 | -0.0031±0.003 |
| 22 | 1.0259±0.005 | 1.0270±0.003 | -0.1060±0.003 | 1.0230±0.003 | 1.0261±0.003 | -0.0030±0.003 |
| 24 | 1.0288±0.006 | 1.0290±0.003 | -0.0183±0.002 | 1.0256±0.004 | 1.0288±0.003 | -0.0031±0.003 |
| 26 | 1.0315±0.004 | 1.0310±0.004 | 0.0472±0.002 | 1.0280±0.004 | 1.0310±0.004 | -0.0029±0.002 |
| 28 | 1.0340±0.005 | 1.0330±0.003 | 0.0939±0.002 | 1.0302±0.005 | 1.0331±0.004 | -0.0028±0.003 |
| 30 | 1.0363±0.005 | 1.0345±0.003 | 0.1728±0.003 | 1.0322±0.004 | 1.0349±0.003 | -0.0026±0.004 |
| 35 | 1.0415±0.004 | 1.0385±0.004 | 0.2873±0.003 | 1.0368±0.004 | 1.0394±0.003 | -0.0025±0.003 |
| 40 | 1.0460±0.006 | 1.0399±0.004 | 0.5871±0.003 | 1.0408±0.004 | 1.0429±0.003 | -0.0020±0.003 |

The buildup-tail model originated from the proportion function $y(x) = \frac{1}{x}$. When $x$ increases from $-\infty$ to 0, the curve of $y$ is located in the region of the (-,-) quadrant. When $x$ goes from 0 to $+\infty$, the curve of $y$ is located in the region of the (+, +) quadrant. Let $\frac{1}{x}$ be $\frac{1}{|x|}$, then the curve of $y(x)$ falls in the (-,+) and the (+,+) quadrants. The curve of $y(x) = \frac{1}{|x|}$ has a left and right tail of the dose-profile-shape pattern. Let $y(x) = \frac{1}{|x|} = \frac{1}{\sqrt{(x^2)}}$. When $x = 0$, $y(x)$ becomes infinite which does not happen in real dose-profiles. Therefore, we insert n into $y(x)$ to be: $f(x) = \left( \frac{1}{\sqrt{n+(x^2)}} \right)$, where n>0., let $tail(x) = \frac{x}{\sqrt{n+x^2}}$, where $x$ is the depth in water in unit of cm, n is a scalar of spread factor in real number.

On other hand, the function $tail(x) = \frac{x}{\sqrt{n+x^2}}$ demonstrates an ascending values $tail(x)$ with an increasing depth of $x$ in water. When introduces an exponential function $e^{\mu x}$ to $tail(x)$, the combination becomes PDD$_{b-t}$ function $\frac{x}{\sqrt{x^2+n}} \cdot e^{-\mu x}$, namely, PDD$_{b-t} = \frac{d}{\sqrt{d^2+n}} \cdot e^{-\mu d}$, where $d$ is the depth in water in the unit of cm, n>0 and is a harden factor of real number scalar. When $x = 0$, n plays an important role to avoid *primary(x)* become infinite, while μ is the linear attenuation factor to fine turn the growth of the $\frac{x}{\sqrt{x^2+n}}$ value.

Finally, the buildup-tail model can be expressed as follows:

$$\mathrm{PDD}_{b-t} = \frac{d}{\sqrt{d^2 + n}} \cdot e^{-\mu d}$$

The PDD can be fitted by using the buildup-tail modeling by adjusting the main parameters of n and μ in all photon energy for the standard PDD curves in Fig 1A–1D. The random

variations of modeled PDD with measured PDD had a maximum deviation within 1.5% as shown in Table 1.

The parameters n and μ represent the photon beam hardening factor of the buildup function and the beam penetration ability of the tail function, respectively. The more photon energy, the more n and less μ it has for the PDD curve fitting as shown in Table 2.

Fig 2 show the parameter n and μ in describing the measured $S_c$ in the energy of 6 and 10 MV photon beam in Fig 2A and 2B, respectively. The $S_c$ of photon energy 6 and 10 MV can be expressed with a certain deviation within 0.8% by using the parameters n and μ generated in empirical buildup-tail function by Eq 2,

Where $n_E$ and $μ_E$ denote the parameters n and μ as a function of field size, FS, in empirical buildup-tail function at photon energy E.

A high energy photon beam usually has a high penetration ability, namely, has a small attenuation coefficient μ and also has a large n to own a less surface dose and the deeper $d_{max}$. The combination of a small μ and a large n, large μ, and small n can characterize a high and low energy photon beam percentage depth dose, respectively (Table 2).

The concept of in-air output ratio $S_c$ was introduced to characterize how the incident photon fluence per monitor unit varies with collimator settings, and the author believe the $S_c$ of a larger photon energy is more sensitive in accordance with the field size, namely, the larger collimator setting, the more $S_c$ it is. This might be owing to the strong side scatter of the higher energy of photons. As we can see in Fig 2B, the $S_c$ of 10 MV increases gradually when the field sizes open widely, on the contrary, Fig 2A, the $S_c$ of 6 MV becomes slightly saturation when the field sizes open widely. This phenomenon causes the curvature of $S_c$ of photon energy 10 MV to be easy fitted by the parameters n and μ as a function of field size than in a photon energy of 6 MV.

Table 3 shows the deviation of characterized and measured $S_c$ of Varian photon energy at 6 and 10 MV.

Since parameter n represents photon beam hardening factor in buildup function, in other words, the larger n (n = 4.95) the high beam quality, therefore you can see the larger n, the less surface dose, and the deeper $d_{max}$ (to compare n = 4.95 and n = 0.0495 in Fig 3).

On the other hand, μ represents the attenuation coefficient, meanwhile, tail function represents the beam penetration ability of high energy photon beam.

As shown in Fig 3, the larger μ (μ = 0.458) the more attenuation when photon penetrating in the medium, therefore, you can see the larger μ, the steeper curves and to have a shortened range (to compare μ = 0.458 and μ = 0.00458).

It can be seen that the change of $S_c$ at square field size with a range of 0.951 to 1.048 for the Elekta 6 MV flattened beam. It is very interesting that a weaker variation can be seen when the upper jaw and lower jaw collimator setting was reversed. For example, the $S_c$ of upper jaw x lower jaw setting at 10 cm x 15 cm and 15 cm x 10 cm was 1.0049, 1.0104 in Table 4, respectively.

The Sc of Elekta 6 MV photon beam with different upper and low jaw setting from 4 cm x 4 cm to 40 cm x 40 cm was listed in Table 4. The orthogonal equal upper and lower jaw setting (square field) from the upper left to the lower right of 4 cm x 4 cm to 40 cm x 40 cm follow the Eq 2 while $S_c$ can be calculated for the different upper jaw and lower jaw setting separately following the equation below,

$$S_c = 0.88 \cdot \left( \left( \sqrt[0.65]{upper\ jaw} \cdot \sqrt[0.35]{lower\ jaw} \right) \right)^{0.06} \qquad \text{Eq3}$$

The $S_c$ of different upper and lower jaw settings can be calculated separately by using Eq 3. In Table 5a, the $S_c$ can be calculated sequentially by using Eq 3 for various lower settings with

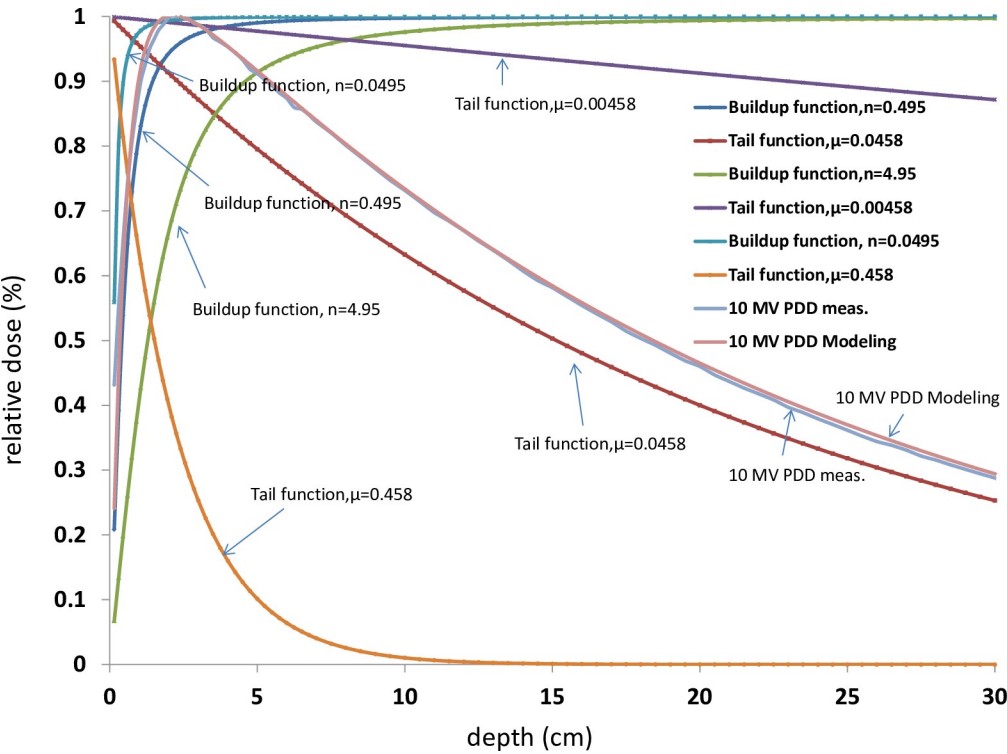

**Fig 3. The parameter n represents the photon beam hardening factor in the buildup, the larger n (n = 4.95) the high beam quality, thus accompany the less surface dose and the deeper d$_{max}$ (to compare n = 4.95 and n = 0.0495 in Fig 2).** The parameter μ represents the attenuation coefficient of the photon beam in the medium. The larger μ (μ = 0.458) the more attenuation, thus, the steeper curves and to have a shortened range (to compare μ = 0.458 and μ = 0.00458).

upper jaw fixed at 4 cm, 5 cm, 20 cm, 30 cm, and 40 cm, while in Table 5b, the S$_c$ was calculated sequentially for various upper setting with lower jaw fixed at 4 cm, 5 cm, 20 cm, 30 cm, and 40 cm. The S$_c$ calculation by separating the individual upper or lower jaws setting does a lot of favor in individual treatment monitor unit double-checking at jaws opened asymmetrically [13].

## V. Conclusions

The dosimetric quantities S$_c$, even the quantity S$_p$ by dividing S$_{cp}$ with S$_c$, and hence required for planning system measurement, or a monitor unit check methodology, were easily and

**Table 4. This figure shows S$_c$ of Elekta 6 MV photon beam with different upper and low jaw setting with field sizes from 4 cm x 4 cm to 40 cm x 40 cm.**

| Upper jaw setting(Y, cm) | Lower jaw setting (X, cm) | | | | | | | |
|---|---|---|---|---|---|---|---|---|
| | 4 | 5 | 10 | 15 | 20 | 25 | 30 | 40 |
| 4 | 0.9511 | 0.9563 | 0.9671 | 0.972 | 0.9739 | 0.9745 | 0.9751 | 0.9752 |
| 5 | 0.9587 | 0.961 | 0.9773 | 0.9796 | 0.9801 | 0.9805 | 0.9814 | 0.9823 |
| 10 | 0.973 | 0.9759 | 1 | 1.0049 | 1.0063 | 1.0077 | 1.0086 | 1.0091 |
| 15 | 0.9798 | 0.9796 | 1.0104 | 1.0163 | 1.0181 | 1.0208 | 1.0222 | 1.0231 |
| 20 | 0.9856 | 0.9891 | 1.0167 | 1.0222 | 1.0258 | 1.0267 | 1.0289 | 1.0308 |
| 25 | 0.9872 | 0.9909 | 1.0204 | 1.0267 | 1.0285 | 1.0308 | 1.0326 | 1.0344 |
| 30 | 0.9903 | 0.9932 | 1.0231 | 1.0289 | 1.0331 | 1.0344 | 1.0376 | 1.0394 |
| 40 | 0.9952 | 0.9955 | 1.0285 | 1.0358 | 1.0381 | 1.0426 | 1.0449 | 1.048 |

**Table 5. a. shows the $S_c$ can be calculated sequentially by using Eq 3 for various lower settings with upper jaw fixed at 4 cm, 5 cm, 20 cm, 30 cm, and 40 cm. b. shows the $S_c$ was calculated sequentially for various upper settings with lower jaw fixed at 4 cm, 5 cm, 20 cm, 30 cm, and 40 cm.**

**a.**

| upper jaw (Y, cm) | lower jaw (X, cm) | $S_c$ (square field) | $S_c$ calculated | error | upper fix at 4 cm, lower jaw increased $S_c$ (4 cm X lower jaw) | $S_c$ calculated | error | upper fix at 5 cm, lower jaw increased $S_c$ (5 cm X lower jaw) | $S_c$ calculated | error | upper fix at 20 cm, lower jaw increased $S_c$ (20 cm X lower jaw) | $S_c$ calculated | error | upper fix at 30 cm, lower jaw increased $S_c$ (30 cm X lower jaw) | $S_c$ calculated | error | upper fix at 40 cm, lower jaw increased $S_c$ (40 cm X lower jaw) | $S_c$ calculated | error |
|---|---|---|---|---|---|---|---|---|---|---|---|---|---|---|---|---|---|---|---|
| 4 | 4 | 0.951 | 0.956 | 0.550 | 0.951 | 0.952 | 0.550 | 0.959 | 0.962 | 0.104 | 0.986 | 0.995 | 0.010 | 0.990 | 0.996 | 0.006 | 0.995 | 1.005 | 0.010 |
| 5 | 5 | 0.961 | 0.965 | 0.384 | 0.956 | 0.955 | 0.384 | 0.961 | 0.964 | -0.173 | 0.989 | 0.998 | 0.009 | 0.993 | 1.000 | 0.007 | 0.996 | 1.009 | 0.013 |
| 10 | 10 | 1.000 | 0.991 | -0.888 | 0.967 | 0.963 | -0.888 | 0.977 | 0.972 | -0.463 | 1.017 | 1.006 | -0.010 | 1.023 | 1.011 | -0.011 | 1.029 | 1.020 | -0.008 |
| 15 | 15 | 1.016 | 1.007 | -0.923 | 0.972 | 0.967 | -0.923 | 0.980 | 0.977 | -0.482 | 1.022 | 1.011 | -0.011 | 1.029 | 1.018 | -0.011 | 1.036 | 1.027 | -0.009 |
| 20 | 20 | 1.026 | 1.018 | -0.733 | 0.974 | 0.971 | -0.733 | 0.980 | 0.980 | -0.333 | 1.026 | 1.015 | -0.011 | 1.033 | 1.023 | -0.010 | 1.038 | 1.032 | -0.006 |
| 25 | 25 | 1.031 | 1.027 | -0.351 | 0.975 | 0.973 | -0.351 | 0.981 | 0.983 | -0.127 | 1.027 | 1.018 | -0.009 | 1.034 | 1.027 | -0.008 | 1.043 | 1.035 | -0.007 |
| 30 | 30 | 1.038 | 1.035 | -0.298 | 0.975 | 0.975 | -0.298 | 0.981 | 0.985 | 0.030 | 1.029 | 1.020 | -0.009 | 1.038 | 1.030 | -0.008 | 1.045 | 1.038 | -0.006 |
| 40 | 40 | 1.048 | 1.046 | -0.174 | 0.975 | 0.979 | -0.174 | 0.982 | 0.989 | 0.366 | 1.031 | 1.023 | -0.007 | 1.039 | 1.034 | -0.005 | 1.048 | 1.043 | -0.004 |

**b.**

| upper jaw (Y, cm) | lower jaw (X, cm) | $S_c$ (square field) | $S_c$ calculated | error | lower fix at 4 cm, upper jaw increased $S_c$ (upper jaw x 4 cm) | $S_c$ calculated | error (%) | lower fix at 5 cm, upper jaw increased $S_c$ (upper jaw x 5 cm) | $S_c$ calculated | error (%) | lower fix at 20 cm, upper jaw increased $S_c$ (upper jaw x 20 cm) | $S_c$ calculated | error (%) | lower fix at 30 cm, upper jaw increased $S_c$ (upper jaw x 30 cm) | $S_c$ calculated | error (%) | lower fix at 40 cm, upper jaw increased $S_c$ (upper jaw x 40 cm) | $S_c$ calculated | error (%) |
|---|---|---|---|---|---|---|---|---|---|---|---|---|---|---|---|---|---|---|---|
| 4 | 4 | 0.951 | 0.956 | 0.550 | 0.951 | 0.940 | -1.164 | 0.956 | 0.946 | -1.042 | 0.974 | 0.987 | 1.297 | 0.975 | 0.981 | 0.634 | 0.975 | 0.982 | 0.699 |
| 5 | 5 | 0.961 | 0.965 | 0.384 | 0.959 | 0.946 | -1.289 | 0.961 | 0.953 | -0.864 | 0.980 | 0.993 | 1.332 | 0.981 | 0.988 | 0.660 | 0.982 | 0.988 | 0.530 |
| 10 | 10 | 1.000 | 0.991 | -0.888 | 0.973 | 0.966 | -0.696 | 0.976 | 0.973 | -0.326 | 1.006 | 1.014 | 0.767 | 1.009 | 1.009 | 0.003 | 1.005 | 1.005 | -0.429 |
| 15 | 15 | 1.016 | 1.007 | -0.923 | 0.980 | 0.978 | -0.179 | 0.980 | 0.985 | 0.512 | 1.018 | 1.026 | 0.818 | 1.022 | 1.021 | -0.120 | 1.023 | 1.015 | -0.791 |
| 20 | 20 | 1.026 | 1.018 | -0.733 | 0.986 | 0.987 | 0.094 | 0.989 | 0.993 | 0.410 | 1.026 | 1.035 | 0.929 | 1.029 | 1.030 | 0.090 | 1.031 | 1.022 | -0.821 |
| 25 | 25 | 1.031 | 1.027 | -0.351 | 0.987 | 0.993 | 0.603 | 0.991 | 1.000 | 0.901 | 1.029 | 1.042 | 1.340 | 1.033 | 1.037 | 0.401 | 1.034 | 1.028 | -0.613 |
| 30 | 30 | 1.038 | 1.035 | -0.298 | 0.990 | 0.999 | 0.8381 | 0.993 | 1.005 | 1.219 | 1.033 | 1.048 | 1.442 | 1.038 | 1.042 | 0.465 | 1.039 | 1.033 | -0.640 |
| 40 | 40 | 1.048 | 1.046 | -0.174 | 0.995 | 1.007 | 1.2114 | 0.996 | 1.014 | 1.860 | 1.038 | 1.057 | 1.829 | 1.045 | 1.051 | 0.628 | 1.048 | 1.040 | -0.744 |

accurately parameterized for our flattened beams using a simple mathematical expression in this study. The data reproduced may be used as expectation values for comparison when commissioning similar beams, as there are scant published data on $S_c$ of all photon beams from these accelerator types.

In this study, we presented a study with an empirical method to model the PDD curve for high energy photon beam by using the buildup and tail function in radiation therapy. The modeling parameters n and μ can also be used to predict the $S_c$ in either square field or with jaws opening asymmetrically for individual treatment monitor unit double check in the patient's dose calculation.

The achievement of this study provides a lot of help in double check of the measurement results to reduce the time of measurements, and the confidence to use interpolation. It would be helpful to know the smallest number of measurements needed to characterize the high-energy x-ray beams.

## Supporting information

**S1 Data.**
(XLSX)

## Acknowledgments

The author appreciates Professor Yan-Cheng Ye for his great contribution to this study. Professor Yan-Cheng Ye is juxtaposed with first author (co-first authors with equal contribution). Professor Yan-Shan Zhang is juxtaposed with correspondence author (co-corresponding authors with equal contribution).

## Author Contributions

**Conceptualization:** Xiao-Jun Li, Jia-Ming Wu.

**Data curation:** Xiao-Jun Li, Yan-Shan Zhang, Jia-Ming Wu.

**Formal analysis:** Xiao-Jun Li, Yan-Shan Zhang, Jia-Ming Wu.

**Funding acquisition:** Yan-Cheng Ye, Yan-Shan Zhang, Jia-Ming Wu.

**Investigation:** Yan-Cheng Ye, Yan-Shan Zhang, Jia-Ming Wu.

**Methodology:** Yan-Cheng Ye, Yan-Shan Zhang, Jia-Ming Wu.

**Project administration:** Jia-Ming Wu.

**Resources:** Xiao-Jun Li.

**Software:** Xiao-Jun Li, Jia-Ming Wu.

**Supervision:** Yan-Cheng Ye, Jia-Ming Wu.

**Validation:** Xiao-Jun Li, Yan-Cheng Ye, Jia-Ming Wu.

**Visualization:** Jia-Ming Wu.

**Writing – original draft:** Jia-Ming Wu.

**Writing – review & editing:** Jia-Ming Wu.

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
