## [Decision Letter · Decision Letter 0]

5 Oct 2021

PONE-D-21-29815Empirical Modeling of the Percent Depth Dose for Megavoltage Photon BeamsPLOS ONE

Dear Dr. Wu,

Thank you for submitting your manuscript to PLOS ONE. After careful consideration, we feel that it has merit but does not fully meet PLOS ONE’s publication criteria as it currently stands. Therefore, we invite you to submit a revised version of the manuscript that addresses the points raised during the review process. Please submit your revised manuscript by Nov 19 2021 11:59PM. If you will need more time than this to complete your revisions, please reply to this message or contact the journal office at plosone@plos.org. Please include the following items when submitting your revised manuscript:A rebuttal letter that responds to each point raised by the academic editor and reviewer(s). You should upload this letter as a separate file labeled 'Response to Reviewers'.A marked-up copy of your manuscript that highlights changes made to the original version. You should upload this as a separate file labeled 'Revised Manuscript with Track Changes'.An unmarked version of your revised paper without tracked changes. You should upload this as a separate file labeled 'Manuscript'.

We look forward to receiving your revised manuscript.

Kind regards,

Suhairul Hashim, PhD

Academic Editor

PLOS ONE

Journal Requirements:

"This work was supported by the Funding: Key R&D plan of Science and Technology Program of Gansu Province, China. (18YF1FH153)."

"This work was supported by the Funding: Key R&D plan of Science and Technology Program of Gansu Province, China. (18YF1FH153)."

Additional Editor Comments:

Please address the following concerns by the reviewers to improvise the current manuscript.

Reviewers' comments:

Reviewer's Responses to Questions

**Comments to the Author**

1. Is the manuscript technically sound, and do the data support the conclusions?

Reviewer #1: Yes

Reviewer #2: Yes

2. Has the statistical analysis been performed appropriately and rigorously? 

Reviewer #1: Yes

Reviewer #2: Yes

3. Have the authors made all data underlying the findings in their manuscript fully available?

Reviewer #1: Yes

Reviewer #2: No

4. Is the manuscript presented in an intelligible fashion and written in standard English?

Reviewer #1: No

Reviewer #2: Yes

5. Review Comments to the Author

Reviewer #1: This manuscript reports the development of an empirical method to model the high-energy photon beam percent depth dose (PDD) curve. It is an interesting and potentially beneficial for generating physical parameters used for monitor unit checking in radiation therapy. The accuracy of the mathematical formulation produced is very impressive.

The manuscript in general is well written. The issue with the manuscript however, is with regard to the formatting and to a lesser extent, the language used. The authors should be more critical with the use of acronyms, and font formatting. For example:

• The acronym PDD should be used after it has been defined when it first appeared in the manuscript.

• Letter ‘c’ for Sc should be subscripted. ‘Sc’ should be used instead of ‘collimator scatter’ after it has been defined.

• Scalar quantity ‘n’ should be italised.

Some of the results are presented in the ‘Discussion’ section, which should go under the ‘Results’ section. More discussions on the methods and comparison with other methods/studies are desirable. The manuscript can also be benefited with proofreading exercise to polish up the language. Some other specific comments as follow:

Abstract:

Page 3, line 54-55. Please acronymise percentage depth dose. Please check throughout the manuscript.

Page 3, line 57. Remove ‘collimator scatter’. Sc has been defined earlier. Please check throughout the manuscript.

Page 4, line 73. Insert ‘to’ between ‘sizes’ and ‘within’.

Page 4, line 77. Add ‘setting’ after ‘clinical’.

Page 4, line 77. Please subscript the letter ‘c’ in Sc. Check throughout manuscript.

Introduction:

Page 6, line 100. Do not capitalise letter ‘m’ in measurement.

Page 6, line 103. Change ‘on’ to ‘of’

Page 6 and 7, line 108-119. The authors presented the physics and a number of physical parameters of high energy x-ray which can be found in common radiation physics textbook, which I strongly feel is unnecessary. The authors should instead provide and explain the problem statement/study background or brief literature review of similar study, leading to the aims/objectives of the manuscript.

Page 7, line 127-133. The detailed explanation on the mathematical formula is not required here, as it is provided in the method. I suggest to just include the objective(s) of the current study, and its importance.

Method:

Page 9, line 159. Change ‘Percent depth curve measurement’ to ‘Percent depth dose measurement’

Page 9, Line 163. Superscript number ‘3’ on cm3.

Page 10, line 181 to 183. Please clarify the statement ‘This was performed by giving each film a priming dose of 2 Gy, to homogenize the film density using WHICH facility with a dose of 1 Gy at the photon energy of 6 MV’. I don’t understand what is mean by priming the film to 2 Gy, with a dose of 1 Gy?

Page 10, line 190. Please specify the version and manufacturer of the software used.

Page 10, line 191. Gafchromic film is specified as GAF film in some places, and back to Gafchromic at some places. Please be consistent.

The explanation of measurement using Gafchromic film in my opinion, is excessive. The method used for calibrating the film is widely published and the authors can just cite the reference(s), instead of explaining the procedures in great detail.

Results:

Page 14, line 246-247. It was stated that the measurement using ion chamber coincide with the film measurement. Please provide quantitative values on the agreement.

Page 14, line 249. Change ‘at the commissioning’ to ‘during commissioning’.

Page 15-16, Figure 1(a)-(d). Despite the excellent agreement between the fitting and measurement, it is imperative to provide the values of goodness of fit (or other quantitate deviation measurement) as a quantitative measure on the fitting performance.

Page 17, line 305-309. The expression of Sc should be provided in the methods section. Is ‘FS’ the dimension of field size? This should be stated.

Page 18, caption for Figure 2. It was stated that ‘the Sc of photon energy 6 MV (a) and 10 MV 321 (b) can be expressed perfectly by using the parameters n and μ’. I disagree to that because albeit from a reasonably good fit, it is not perfect. There are some slight disagreements at certain field sizes.

Page 19, Table 2. I noted the deviations for 6 MV are in general larger than 10 MV. Any explanations for this?

Discussion:

Some of the results presented here should go under the ‘Results’ section. For instance, page 21, lines

Page 21, lines 383-390.

Page 24, line 440. Change ‘interest’ to ‘interesting’.

Page 24, line 440-442. It was stated that ‘It is very interest that a weaker variation can be seen when the upper jaw and lower jaw collimator setting was reversed’. Any plausible reason/explanation for this?

Some tables here (Figure 3 and 4) are results and should go under the results section.

More discussion on the utility of the empiral mathematical formula is desirable. Are there any similar empirical methods/mathematical formula reported in the literature? What are the benefits of this method for generating physical parameters for MU checking? What is the future work/recommendations as I believe more work is required to verified the accuracy/feasibility of using such formula in clinical setting?

Reviewer #2: Title: Empirical Modeling of the Percent Depth Dose for Megavoltage Photon Beams

Manuscript number: PONE-D-21-29815

General comments:

This paper describes and empirical method to model the PDD of the high-energy photon beam using a mathematical – buildup-tail function. The study was interesting because the proposed mathematical model could be useful to predict the collimator scatter factor, Sc for PDDs.

Please clarify if the measured PDD in Fig 1a-d, were PDD from published literature or from ion chamber measurements. If they were from published data, how did the author managed to acquire such detail of individual measurement points?

There are numerous language and grammatical errors throughout the manuscript, unnecessary capitalization at random places, subscripts for Sc etc should be carefully pick-up and corrected by the authors.

Specific comments:

Introduction – Could the authors present the background and the need for this study? Because, typically PDDs and Sc are measured from the actual beam, because the beams are different for every linac, and depending on the detector used to measure it.

Line 100 – measurement – do not capitalize “M”

Line 149 and many other places – Sc - to make sure the c is subscripted

Line 153 – four photon energies

Line 159 – Percent depth dose curve

Line 163 – please specify the company, state and country of the PTW. The volume is cm3.

Line 182 – What is the reason to prime the film for 2Gy? Was it 1Gy or 2Gy? Please clarify? If the films are primed with 1-2 Gy, then, how would that affect the calibration curves? Where is the 0 Gy point?

Line 186 – Why is the gafchromic calibration curve called a Hurter-Diffield calibration curve?

There were description of the use of gaf film – but I fail to see the purpose of the film measurements, since the authors only used the ion chamber measured data.

Line 200 – 214 – Please explain the linac used to measure the Sc.

Line 222 – no need to capitalize percent depth dose

Line 237 – four photon energies

Line 240 – please write in full sentence.

Figure 1a – d – The two modeled and published PDD appears to match very well, but, at the region of dose buildup and end of tail, the dose descrepencies are obvious. Perhaps an additional graph on the error/difference between the 2 curves can be added to the figures to show the differences?

Figure 2 a and b – again, what are the errors of the measured values, i.e. error bars? Where were these values obtained from?

Table 1 – These parameters are specific for the 4 PDDs that the authors had modeled on. Please discuss how these values are going to be applicable for any PDDs?

Line 306 – no need to capitalize buildup

Table 2 – shows the error between the measured and predicted Sc. Perhaps, displaying this in a graph format would be better and summarize the average, SD or max deviation at what field size. Unless the numerical value has any value for any PDD or field sizes of the reader.

Line 359 – no need to capitalize buildup

Interesting explanation of the equations. However, one of the limitation of the model, correct me if I am wrong, is unable to predict the dose at the surface, at d = 0 cm and the discrepancy of the model at the dose buildup region could be quite big. Can the authors comment on the errors at these regions?

Line 427 – no figure 3 caption.

Table 3 – should this be in the results?

6. PLOS authors have the option to publish the peer review history of their article (what does this mean?). If published, this will include your full peer review and any attached files.

Reviewer #1: No

Reviewer #2: No

---

## [Author Response · Author response to Decision Letter 0]

14 Oct 2021

Please check the point-to-point respond for the reviewer's file, and the color highlighted file for further procession. Thank you so much.

---

## [Decision Letter · Decision Letter 1]

23 Nov 2021

Empirical Modeling of the Percent Depth Dose for Megavoltage Photon Beams

PONE-D-21-29815R1

Dear Dr. Wu,

We’re pleased to inform you that your manuscript has been judged scientifically suitable for publication and will be formally accepted for publication once it meets all outstanding technical requirements.

Kind regards,

Suhairul Hashim, PhD

Academic Editor

PLOS ONE

Additional Editor Comments (optional):

Reviewers' comments:

Reviewer's Responses to Questions

**Comments to the Author**

1. If the authors have adequately addressed your comments raised in a previous round of review and you feel that this manuscript is now acceptable for publication, you may indicate that here to bypass the “Comments to the Author” section, enter your conflict of interest statement in the “Confidential to Editor” section, and submit your "Accept" recommendation.

Reviewer #1: All comments have been addressed

Reviewer #2: All comments have been addressed

2. Is the manuscript technically sound, and do the data support the conclusions?

Reviewer #1: Yes

Reviewer #2: Yes

3. Has the statistical analysis been performed appropriately and rigorously? 

Reviewer #1: Yes

Reviewer #2: Yes

4. Have the authors made all data underlying the findings in their manuscript fully available?

Reviewer #1: Yes

Reviewer #2: Yes

5. Is the manuscript presented in an intelligible fashion and written in standard English?

Reviewer #1: Yes

Reviewer #2: Yes

6. Review Comments to the Author

Reviewer #1: The authors have addressed my earlier concerns. I can now recommend to accept the paper for publication.

Reviewer #2: The authors has adequately address the questions and the manuscript is much improved. I recommend for the acceptance of the manuscript.

7. PLOS authors have the option to publish the peer review history of their article (what does this mean?). If published, this will include your full peer review and any attached files.

Reviewer #1: No

Reviewer #2: No

---

## [Editor Report · Acceptance letter]

20 Dec 2021

PONE-D-21-29815R1 

Empirical Modeling of the Percent Depth Dose for Megavoltage Photon Beams 

Dear Dr. Wu:

I'm pleased to inform you that your manuscript has been deemed suitable for publication in PLOS ONE. Congratulations! Your manuscript is now with our production department. 

Kind regards, 

on behalf of

Dr. Suhairul Hashim 

Academic Editor

PLOS ONE